# Flowers of *Astragalus membranaceus* var. *mongholicus* as a Novel High Potential By-Product: Phytochemical Characterization and Antioxidant Activity

**DOI:** 10.3390/molecules24030434

**Published:** 2019-01-25

**Authors:** Yuan Li, Sheng Guo, Yue Zhu, Hui Yan, Da-wei Qian, Han-qing Wang, Jian-qiang Yu, Jin-ao Duan

**Affiliations:** 1Jiangsu Collaborative Innovation Center of Chinese Medicinal Resources Industrialization, State Administration of Traditional Chinese Medicine Key Laboratory of Chinese Medicinal Resources Recycling Utilization, National and Local Collaborative Engineering Center of Chinese Medicinal Resources Industrialization and Formulae Innovative Medicine, Nanjing University of Chinese Medicine, Nanjing 210023, China; meredithtly@163.com (Y.L.); nzyzy808@163.com (Y.Z.); glory-yan@163.com (H.Y.); qiandwnj@126.com (D.-w.Q.); 2School of Pharmacy, Ningxia Medical University, Yinchuan 750021, China; wwwhhq@163.com (H.-q.W.); yujq910315@163.com (J.-q.Y.)

**Keywords:** *Astragalus membranaceus* var. *mongholicus*, antioxidant activity, flavonoid, phenolic acid

## Abstract

The root of *Astragalus membranaceus* var. *mongholicus* is one of the most popular herbal medicines worldwide. In order to increase the yield of underground roots of *A.*
*membranaceus* var. *mongholicus*, its flowers (AMF) have often been removed in their flowering stage, which produces the flowers as waste being discarded. To explore its phytochemicals and potential value for utilization, the antioxidant activities of extracts from AMF were evaluated by a free radical scavenging assay and reducing power assay. The total phenols and flavonoids, as well as the individual compounds, in different extracts of AMF were also investigated. The results showed that the extract ME obtained from AMF through macroporous resins separation exhibited strong antioxidant activities, which were close to those of positive control BHT. ME was rich in phenolic acids and flavonoids, and the contents reached 108.42 mg gallic acid equivalents/g and 265.70 mg rutin equivalents/g, respectively. A total of 31 compounds, including four phenolic acids, nineteen flavonoids, three isoflavones, two pterocarpans, and three saponins, were identified using UPLC-QTOF-MS in ME. Quantitative analysis of sixteen components in the extracts of AMF showed that flavonoids were the predominant constituents, especially for the compounds of hyperoside, rutin, and isorhamnetin-3-*O*-β-d-glucoside.

## 1. Introduction

*Astragalus membranaceus* and its variety *A. membranaceus* var. *mongholicus*, as the perennial herbs in the *Leguminosae* family, are mainly distributed in Northeast, North, and Northwest China, as well as in Mongolia and Korea [1]. Their root (Astragali Radix), known as Huangqi in China, is one of the most popular herbal medicines worldwide, with immunomodulating [2,3] antihyperglycemic [4], antiinflammatory [5], antioxidant [6,7], and antiviral activities. As one of the most important Qi tonifying adaptogenic herbs in Traditional Chinese Medicine, Astragali Radix has been utilized as a crude drug, as well as food, in China for thousands of years [8].

In recent years, owing to the increasing demand for its use as a raw material for health food, the natural resources of Astragali Radix have been diminishing. At present, the resources of Astragali Radix mainly originate from cultivated plants in China [9]. In order to increase the yield of its underground roots, the flowers of *A. membranaceus* and *A. membranaceus* var. *mongholicus* (AMF) are often removed in their flowering stages, which produces large amounts of the flowers as waste being discarded and has caused a great waste of plant resources [10,11]. Thus, it is necessary to transform the flower waste of the processed crop product into ecologically friendly or sustainable material suitable for industrial purposes, in particular, for health products that are in high demand.

According to previous studies, the flowers of some species in *Astragalus* genus are reported to contain a relatively high amount of flavonoids, triterpenoid saponins, phenolic acids, and volatile compounds [12,13,14]. Besides, flavonoids, sugars, amino acids, and triterpenoid saponins were definitely detected in the flower of *A. membranaceus* [15,16,17]. Owing to the activities of the above compounds previously reported, AMF could be a promising raw material of health products. However, the available information on its bioactivity is rather scarce.

It is well-known that free radicals are considered to be responsible for damage to lipids, proteins, and nucleic acids in cells, which play a key role in the pathogenesis of various diseases [18]. Thus, exploring the natural antioxidants for use in pharmaceutical or food products has gradually attracted attention in recent years in order to limit the use of synthetic antioxidants, due to side effects. As a Chinese tonic medicine, the roots of *A. membranaceus* and *A. membranaceus* var. *mongholicus* have shown obvious preventive effects on tissue injury via antioxidant mechanisms [19], while little information about the antioxidant activity was found for their flowers.

Accordingly, the purpose of this study was to verify the antioxidant activity and its related chemical constituents in AMF. Their antioxidant activities were investigated by a free radical scavenging assay and reducing power assay. The total phenols and flavonoids, as well as the individual compounds, in the extracts of AMF were also investigated, and the results can demonstrate the possibility of utilizing the AMF as a novel by-product in pharmaceutical, food, or personal care applications.

## 2. Results and Discussion

### 2.1. Antioxidant Activity of the Extracts from AMF

To screen the potential activity of AMF, the antioxidant capacity of the samples was determined using DPPH and ABTS radical scavenging activity, as well as FRAP assays. The results are presented in Figure 1. As shown, the antioxidant capacity of EE (ethanol extract) was significantly higher than that of AE (aqueous extract). Further separation of EE was performed on an AB-8 macroporous resin column, and the three chemical fractions LE (freeze-dried powder of 10% eluent), ME (freeze-dried powder of 50% eluent), and HE (freeze-dried powder of 90% eluent) were obtained with different ratios (10%, 50%, and 90%, *v/v*) of ethanol in water, and their antioxidant activities were evaluated.

As shown in Figure 1A,B, the ability of AMF extracts to scavenge free ABTS and DPPH radicals showed a growing trend with increasing sample concentration. The ME fraction showed strong DPPH radical-scavenging activity due to its excellent IC_50_ value (35.10 μg/mL) compared to that of BHT (25.28 ± 0.59 μg/mL) (Figure 1C). The results (Figure 1C) of the ABTS test also showed that ME exhibited the best value (22.02 μg/mL) compared to the IC_50_ value of BHT (17.03 ± 0.75 μg/mL). Compared to ME, both LE and HE displayed lower scavenging activity of DPPH and ABTS free radicals, with the IC_50_ values all being higher than 150 μg/mL. In the FRAP assay (Figure 1D), most of the fractions exhibited dose-dependent Fe^3+^ reducing power, and ME, once again, exhibited the highest reducing ability (3.00 ± 0.13 μmol Fe^2+^/g), which was close to that of positive control BHT (5.68 ± 0.62 μmol Fe^2+^/g), while only 0.41 ± 0.02 μmol Fe^2+^/g and 0.78 ± 0.03 μmol Fe^2+^/g were found for LE and HE, respectively.

On the basis of the above findings, the chemical fraction of ME from AMF seems to be attractive as an important source of antioxidants for the food and pharmaceutical industries.

### 2.2. Extraction Yields, Total Phenols, and Flavonoids Contents

It is known that phenols and flavonoids, as major groups of polyphenolic compounds, have a strong relationship with the antioxidant activities of medicinal plants [20]. Thus, the contents of total phenols and flavonoids in AMF extracts, including EE, LE, ME, and HE, were determined by spectroscopic methods, and the results are presented in Figure 2. The TPC values for the four fractions range from 12.29 ± 0.52 mg to 108.42 ± 0.89 mg chlorogenic acid equivalents (CAE)/g dried extract, while their TFC values range from 6.58 ± 0.39 mg to 265.71 ± 21.51 mg rutin equivalents (RE)/g dried extract. The highest TPC and TFC values were observed for the ME fraction, which was in agreement with the results of the antioxidant assay, indicating that there could be a positive correlation between them.

### 2.3. Characterization of Compounds in AMF Extract by UPLC-QTOF-MS

The characterization of the compounds, as performed by UPLC-DAD/QTOF-MS analysis, was only conducted in the ME extract, owing to its best results in the antioxidant activity assays. The detected and identified compounds are listed in Table 1, together with the corresponding retention time, UV λ_max_, pseudo-molecular ion, and main fragment ions in MS^2^. The total ion chromatograms obtained from the positive (A) and negative (B) ions are shown in Figure 3. Based on the present UPLC separation and the evaluation of the MS/MS spectra, ME was found to contain a great variety of polyphenolic compounds, including phenolic acids, flavonoids, isoflavones, and pterocarpans, as well as other unidentified compounds with defined UV λ_max_ and MS fragment characteristics. Fifty-one peaks have been detected by removing the impurity and low signal-to-noise ratio peaks, and 31 compounds among them have been characterized and identified, and their structures are presented in Appendix A.

#### 2.3.1. Phenolic Acids

Peaks **1**, **2**, **3**, and **9** all exhibited the typical fragment ion of [M − H − CO_2_], as well as the UV λ_max_ at 290–320 nm, suggesting that they could be the phenolic acids. According to the literature [21,22] and reference products, peaks **1** and **2** were identified as benzoic acid derivative vanillic acid and protochatechuic acid, respectively. Peak **3** was identified as caffeic acid by comparison with the reference compound. Peak **9** showed a quasi-molecular ion at *m*/*z* 193.0515 ([C_10_H_9_O_4_]^−^), which further generated fragment ions at *m*/*z* 178.0311 ([C_9_H_6_O_4_]^−^) and 134.0392 ([C_8_H_6_O_2_]^−^) by losing CH_3_ or both CH_3_ and CO_2_ groups. Unequivocally, it was identified as ferulic according to the reference [23] and standard compound.

#### 2.3.2. Flavonoids

Peaks **6**, **7**, **8**, **11**, **12**, **15**, **22**, and **25** all exhibited the typical UV profile of flavonol with the λ_max_ value ranging between 255 and 350 nm, and the aglycone ion at *m*/*z* 303 and product ion at *m*/*z* 287 were also found in their MS/MS spectrum. These characteristic ion rules were very useful for screening flavonoids, and the flavone aglycone (**25**) was characterized as quercetin by comparison with the reference compound. Neutral losses of [M + H − rha]^+^ (*m*/*z* 465.0987) and [M + H − glc − rha]^+^ (*m*/*z* 303.0524) were exhibited for peak **7**, which was identified as rutin by comparing it to the reference compound and the literature [24]. Peaks **11** and **12** all presented the quasi-molecular ion at *m*/*z* 465.10 and the neutral loss of [M + H − 162]^+^ (loss of hexosyl unit), and they were characterized as hyperoside and isoquercitrin, respectively, according to the previous reports [25,26] and reference compounds. Peak **6** and **8** all showed similar quasi-molecular ions at *m*/*z* 597.1486 [M + H]^+^ and the fragment ions of [M + H − 132]^+^ (*m*/*z* 465.1012) and [M + H − 132 − 162]^+^ (303.0500) by the losses of the pentosyl unit or both pentosyl and hexosyl units, respectively. However, the exact nature of the sugars and the position of linkages between glycosides could not be ascertained. Thus, these two compounds were only tentatively assigned as quercetin-*O*-hexoside-pentoside. Similarly, peak **22** was tentatively identified as quercetin-*O*-hexoside- rutinoside due to the presence of the ions of [M + H]^+^ (*m*/*z* 773.2129), [M + H − 162]^+^ (*m*/*z* 611.6411), [M + H − 162 − 146]^+^ (*m*/*z* 465.1052), and [M + H − 162 − 146 − 162]^+^ (*m*/*z* 303.0500). Peak **15** yielded a quasi-molecular ion [M − H]^−^ at *m*/*z* 549.0860, releasing an MS^2^ fragment at *m*/*z* 301 [M − H − 162 − 86]^−^, which might correspond to the loss of a malonyl-hexoside moiety. Therefore, peak **15** was tentatively assigned as quercetin-*O*-malonyl-hexoside.

Peaks **13**, **14**, **17**, **27**, and **36** all produced positive ions at *m*/*z* 317 and *m*/*z* 303, which were similar to those generated by isorhamnetin, according to the literature [27]. Thus, peak **36** could be tentatively identified as isorhamnetin, and the other peaks could be assigned as the glycosylated derivatives of isorhamnetin, owing to the neutral losses of sugar moieties being observed in their MS^2^ spectrum. For peak **13**, the pseudo-molecular ion peak at *m*/*z* 647.1636 [M + Na]^+^ and fragment ions of [M + H − rha]^+^ (*m*/*z* 479.1188) were found, and it was identified as isorhamnetin-3-*O*-rutinoside by comparing mass spectra with data from the literature [27]. Similarly, peak **17** was identified as isorhamnetin-3-*O*-β-d-glucoside owing to the presence of [M + H − 162]^+^, which has been confirmed by the reference compound. Considering the common fragments at *m*/*z* 317 ([isorhamnetin + H]^+^) and *m*/*z* 479 ([isorhamnetin + H + 162]^+^) generated by both peaks **14** and **27**, peak **27** was tentatively assigned as isorhamnetin-*O*-hexoside due to the presence of the quasi-molecular ion of [M + H]^+^ at *m*/*z* 479.1185, and peak **14** was assigned as an isorhamnetin-*O*-hexoside derivative.

Peak **16** ([M + H]^+^ at *m*/*z* 449) released a fragment at *m*/*z* 287 [M + H − 162]^+^, which was similar to that of compound **33**, suggesting that compound **16** could be a glycosylated derivative of compound **33**. By comparison with the commercial standards, peaks **16** and **33** were identified as astragalin and kaempferol, respectively.

Peaks **30**, **31**, and **34** were assigned as glycosylated derivatives of rhamnocitrin, according to the common fragment at *m*/*z* 301.07 [rhamnocitrin + H]^+^, as well as the UV profile with peak **49**, which was identified as rhamnocitrin with the reference standard. The compound corresponding to peak **31** was hypothesized as rhamnocitrin-*O*-hexoside, considering its pseudo-molecular ion at *m*/*z* 463 and the fragment at *m*/*z* 301 ([M + H − hexosyl]^+^). Peak **34** exhibited the pseudo-molecular ion at *m*/*z* 549, which suggested the presence of a malonylhexoside moiety ([rhamnocitrin + H + 162 + 86]^+^), thus the compound corresponding to peak **31** was hypothesized as rhamnocitrin-*O*-hexoside-malonate. Peak **30**, in turn, should correspond to rhamnocitrin-malonylhexoside-rhamnoside, considering the pseudo-molecular ion at *m*/*z* 695 and the fragment at *m*/*z* 549 ([M + H − 146]^+^), *m*/*z* 463 ([M + H − 146 − 86]^+^), and *m*/*z* 301([M + H − 146 − 86 − 162]^+^). All these data were consistent with those reported in the literature [28].

#### 2.3.3. Isoflavones

Compound **10** yielded a [M + H]^+^ ion at *m*/*z* 447.1312 and a major product ion at *m*/*z* 285.0738, and showed a loss of 162 Da, which was deduced to occur via the loss of a glucoside moiety. It was unambiguously identified as calycosin-7-*O*-β-d-glucoside, which was confirmed by the comparison of its t_R_ and mass data with the reference standard. Compound **26** ([M + H]^+^ at *m*/*z* 285.0756) could release the ion at *m*/*z* 270.0500 as the predominant fragmentation due to the loss of a methyl unit. Finally, this compound was identified as calycosin by comparing its mass spectra with the data of the literature [29] and reference compound. The MS spectrum of peak **19** showed [M + H]^+^ at *m*/*z* 463.1234 and fragment ions of [M + H − glc]^+^ at *m*/*z* 301.0672 and [M + H − glc − CH_3_]^+^ at *m*/*z* 286.1731. It was identified as pratensein-7-*O*-β-d-glucoside by comparing the MS data with the literature [28].

#### 2.3.4. Pterocarpans

The [M + H]^+^ ion of compound **28** at *m*/*z* 463.1592 produced the aglycone ion at *m*/*z* 301.0672 in the MS^2^ spectrum, which originated from the neutral loss of a glucose moiety (162 Da). This compound was unambiguously identified as (−)-methylinissolin 3-*O*-β-d-glucoside (9,10-diMP-3-*O*-glucoside) by comparing its t_R_ values and mass spectra with the data of the literature [28] and reference standard. Compound **35** displayed a similar pattern with compound **28**, despite an additional loss of 42 Da, suggesting the presence of an acetyl-glucose moiety. Thus, this compound was tentatively assigned as 9, 10-diMP-3-*O*-acetyl-glucoside.

#### 2.3.5. Saponins

Peak **38** produced the typical ions of *m*/*z* 827.4484 [M + H]^+^ and *m*/*z* 825.4608 [M − H]^−^, which indicated that its molecular weight was 826 (C_43_H_70_O_15_). In addition, *m*/*z* 647.3793 [M + H − glc]^+^ and *m*/*z* 629.3701 [M + H − glc − H_2_O]^+^ were also found. The compound was identified as isoastragaloside II by comparing the MS data of literature [30] and reference compound. Compound **47** (*m*/*z* 913.5197 [M + H]^+^) exhibited a similar pattern with compound **38**, despite an additional loss of 86 Da, indicating the presence of a malonyl unit. Thus, this compound was tentatively deduced as a malonylastragaloside II isomer.

Compounds **48** showed a molecular ion [M + Na]^+^ and [M + H]^+^ at *m*/*z* 965.5129 and 943.5306, respectively. Other important fragments in the spectrum were at *m*/*z* 97.4725 [M + H − rha]^+^, *m*/*z* 635.4149 [M + H − glc − rha]^+^, *m*/*z* 617.4098 [M + H − glc − rha − H_2_O]^+^, and *m*/*z* 599.3946 [M + H − glc − rha − 2H_2_O]^+^. This compound was deduced as soyasaponin I according to the literature [21], which was confirmed by comparing it with the reference standard.

#### 2.3.6. Unknown Compounds

The structure of the compounds marked as unidentified (**18**, **23**, **24**, **39**, **40**, **41**, **43**, **44**, **45**) has not yet been revealed. Among the unidentified-*O*-hexoside-hexoside (**4**), unidentified-*O*-hexosides (**5**, **32**, **50**, **51**), unidentified-*O*-rhamnosides (**29**, **42**, **46**), and unidentified-*O*-pentoside (**37**), only the presence, type, and the linkage of the sugar moiety could be evidenced.

Some peaks (**19** and **20**) could not be identified currently, while they showed UV absorption in two regions, 240–260 nm and 340–370 nm, indicating the existence of multi-conjugated systems in their chemical constituents [31], which conforms to the characteristics of flavonoid compounds. Thus, we speculate that the above compounds may be flavonoid derivatives.

### 2.4. Quantification of Target Compounds in AMF Extract by UPLC-TQ-MS/MS

To reveal the contents of the phenolic acids and flavonoids compounds in different extracts of AMF, a UPLC-TQ-MS/MS method was established and a total of 16 polyphenols, including four phenolic acids and 12 flavonoids, were simultaneously determined. The validation of the proposed method was performed by determining the linearity, LOD, LOQ, precision, repeatability, stability, and recovery, and the results are shown in Appendix A and Appendix A. All the marker substances showed good linearity, with the determination coefficients (R^2^) ranging from 0.9917 to 0.9999 in a relatively wide concentration range. The results of precision, repeatability, and stability tests of the 16 analytes were less than 4.85%, and the recoveries of the analyzed compounds were 95.31–102.10%, with RSDs less than 4.73%. The above results suggested that the proposed method is accurate, precise, and sensitive enough for a quantitative evaluation of those bioactive components in AMF extracts.

The established UPLC–TQ-MS/MS method was then subsequently applied to simultaneous determination of phenolic acids (protochatechuic acid, caffeic acid, vanillic acid, ferulic acid) and flavonoids (rutin, calycosin-7-*O*-β-d-glucoside, hyperoside, astragalin, isorhamnetin-3-*O*-β-d-glucoside, (−)-methylinissolin-3-*O*-β-d-glucoside, quercetin, calycosin, kaempferol, isorhamnetin, formononetin, rhamnocitrin) in AE, EE, LE, ME, and HE samples of AMF. The results (Table 2) showed that there were remarkable differences among the contents of the 16 target compounds in different extracts of AMF. EE has higher contents of phenolic acids (0.886 ± 0.058 mg/g) and flavonoids (16.973 ± 0.854 mg/g) than AE. ME was found to be the richest of phenolic acids (1.871 ± 0.026 mg/g) and flavonoids (36.399 ± 1.230 mg/g), which were far more than those in LE and HE. As for the individual compounds in ME, hyperoside was found to be the predominant constituent with the content of 16.285 ± 0.195 mg/g, nearly one half of the total content of flavonoids analyzed in the assay, followed by rutin (6.099 ± 0.080 mg/g), isorhamnetin-3-*O*-β-d-glucoside (4.970 ± 0.048 mg/g), and astragalin (4.810 ± 0.028 mg/g). Compared with ME, HE mainly contained the flavone aglycones, such as calycosin, rhamnocitrin, and quercetin with the contents of 4.558 ± 0.073 mg/g, 2.184 ± 0.053 mg/g, and 1.326 ± 0.055 mg/g, respectively. While for LE, only four analytes were detected with a total content of less than 0.5 mg/g. Owing to the fact that the total content of flavonoids was significantly higher than the total content of phenolic acids in AMF extracts, it indicates that the flavonoids with reliable biological activities might be representative and abundant compounds of AMF.

### 2.5. Correlation Matrix Analysis

In order to better appreciate the relationships among the antioxidant capacities, TPC, TFC, and the contents of 16 quantitative compounds, Pearson’s correlation coefficients were calculated and the results are summarized in (Table 3).

Strong relationships (*r* 0.658 to 0.948) [32] were observed between the antioxidant capacity (DPPH, ABTS, and FRAP) and the TPC and TFC in AMF extracts. This finding indicated that the antioxidant activities most probably might be contributed by polyphenols contents in AMF, which was in agreement with the previous reports [33].

As for the individual compounds, the phenolic acids, except caffeic acid, showed strong relationships (*r* from 0.785 to 0.980) with the antioxidant capacities. The flavone glycosides exhibited a higher correlation with the antioxidant capacities than the aglycones. For example, isorhamnetin-3-*O*-β-d-glucoside was highly correlated with DPPH, ABTS, and FRAP (r = 0.672, r = 0.660, r = 0.943, respectively), while the correlation coefficients of its aglycone isorhamnetin with DPPH, ABTS, and FRAP were only −0.190, −0.048, and −0.027, respectively. A similar phenomenon was also found between calycosin-7-*O*-β-d-glucoside and calycosin. However, Pietta [34] demonstrated that the presence of a hydroxyl group in the heterocyclic ring also increases the radical-scavenging activity, while glycosylation greatly reduces the radical-scavenging. The change of antioxidant ability caused by the interaction of many compounds may be one of the reasons for the strong antioxidant ability of glycosides. In addition, the number of phenolic hydroxyl groups in aglycones is also an important index affecting their antioxidant activity, which is consistent with earlier reports [35,36]. This phenomenon was confirmed by the higher correlations of quercetin with antioxidant capacities than those of kaempferol, formononetin, and isorhamnetin.

## 3. Materials and Methods

### 3.1. Chemicals and Materials

Fresh AMF samples were collected from two-year-old cultivation *A. membranaceus* var. *mongholicus* plants in Hunyuan County (Shanxi, China), July 2016, and originally identified by Prof. Jin-ao Duan, Nanjing University of Chinese Medicine. In the local area, AMF was dried in shade, crushed into fine powder (40 mesh), and preserved under −20 °C protection from light for analysis.

Butylated hydroxytoluene (BHT), 2,2-diphenyl-1-picrylhydrazyl (DPPH), 2,2′-azinobis-(3-ethylbenzothiazoline-6-sulfonic acid) (ABTS), 2,4,6-tri (2-pyridyl)-striazine (TPTZ), iron (III) chloride, and Folin-Ciocalteu reagent were purchased from Sigma-Aldrich (city, state abbreviation, USA.). Sodium acetate trihydrate, iron (II) sulfate hepta-hydrate, hydrochlorid acid, and sodium carbonate were from Sinopharm Chemical Reagent Co. Ltd. (Shanghai, China). Phosphate buffer and ethanol were purchased from WuXi ChemicalWorks (WuXi, Jiangsu, China). Acetonitrile and methanol were HPLC-grade from Merck (Darmstadt, Germany). Ultra-pure water was prepared by a Millipore Direct Q5 purification system (Millipore, Bedford, MA, USA). AB-8 macroporous resin was purchased from Solarbio (Beijing, China). All other used chemicals were of analytical grade (Nanjing Chemical Plant, Nanjing, China).

Chemical standards, including protochatechuic acid, caffeic acid, vanillic acid, rutin, calycosin-7-*O*-β-d-glucoside, hyperoside, ferulic acid, astragalin, isorhamnetin-3-*O*-β-d-glucoside, (−)-methylinissolin-3-*O*-β-d-glucoside, quercetin, calycosin, kaempferol, isorhamnetin, formononetin, soyasaponin I, rhamnocitrin, and chlorogenic acid, were purchased from Nanjing LongWave biological science and technology Co. Ltd. (Nanjing, China). The purity of each reference compound was over 98%, as determined by HPLC analysis.

### 3.2. Preparation of Plant Extracts

Three samples of AMF powder (40 mesh), each 300.0 g, were taken and accurately weighed. One of them was extracted with water (3 L, 3 × 60 min) in an ultrasonic bath at room temperature, and the combined filtrates were concentrated under reduced pressure at 65 °C, and concentrated solution was frozen at −80 °C overnight and then freeze-dried using a Labconco FreeZone (5 L, Kansas City, MO, USA). The drying was conducted at 0.04 mbar of vacuum, with a drying temperature programmed from −30 to 25 °C. The drying time was 72 h. As a result, a total of 96.0 g aqueous extract (AE) with a water content less than 10% was obtained. The other sample was extracted with ethanol (3 L, 3 × 60 min) and was subsequently processed as the above procedure to obtain the sample of ethanol extract (EE) of 125.4 g. The last one was extracted with ethanol (3 L, 3 × 60 min) in an ultrasonic bath at room temperature, and the combined filtrates were concentrated to 200 mL under reduced pressure at 55 °C. Then, the ethanol-removed solution was separated by an AB-8 macroporous resin column (7.0 cm × 80 cm) eluted with different ratios (10%, 50%, and 90%, *v*/*v*) of ethanol in water, respectively, until each gradient effluent was colorless. Subsequently, the eluents were condensed and dried using a freeze dryer to obtain the samples of LE (freeze-dried powder of 10% eluent) 60.5 g, ME (freeze-dried powder of 50% eluent) 9.3 g, and HE (freeze-dried powder of 90% eluent) 6.9 g, with a water content of less than 10%, respectively. All the dry extracts were stored at 4 °C until used.

### 3.3. Antioxidant Assays

Each dry extract was dissolved in 80% methanol at a concentration of 2 mg/mL and then diluted to prepare the series of concentrations for antioxidant assays, which were performed on an Enspire multifunctional enzyme labeling instrument (Perkin Elmer, Foster City, CA, USA) at the respective wavelengths. All measurements were run in triplicate. The respective antioxidant capacity parameters were also determined for the reference compound BHT as the positive control. All analyses were realized as much as possible in an area protected against light.

#### 3.3.1. DPPH Radical Scavenging Activity Assay

The DPPH free radical scavenging activity of the AMF extracts was measured according to the method of Dong et al. [37]. The reaction mixture was shaken and incubated in the dark at room temperature for 30 min, and the absorbance was read at 517 nm against the blank. The DPPH radical scavenging percentage (%) was calculated using the formula: [1 − (A_1_ − A_2_)/A_3_] × 100%, where A_1_ is the absorbance of reaction mixture, A_2_ is that of the sample only in the absence of DPPH, and A_3_ is the absorbance of the control reaction (containing all reagents except the test sample).

#### 3.3.2. ABTS Radical Scavenging Assay

The ABTS assay of the fractions from AMF was performed as reported previously [38], and the absorbance was read at 734 nm after 6 min incubation. The ABTS radical scavenging percentage (%) was calculated using the formula: [1 − (A_1_ − A_2_)/A_3_] × 100%, where A_1_ is the absorbance of reaction mixture, A_2_ is that of the sample only in the absence of ABTS solution, and A_3_ is the absorbance of the control reaction only without sample.

#### 3.3.3. Ferric Reducing Antioxidant Potential (FRAP) Assay

The FRAP assay was conducted according to the method in the literature [38], and the absorbance was measured at 593 nm. The FRAP results were expressed as μmol Fe^2+^ equivalents per gram of dried extracts (μmol Fe^2+^/mg). All measurements were done in triplicate.

### 3.4. Total Phenols (TPC) and Flavonoids Contents (TFC) Quantification

TPC was determined by the Foline–Ciocalteu method [39] and the result was expressed as milligrams of chlorogenic acid equivalents per gram of dry weight (mg CAE/g dw). TFC was determined by an aluminum nitrate colorimetric assay using rutin as a reference compound, as described in the literature [40], and the result was expressed as milligrams of rutin equivalents per g of dried fraction (mg RE/g dw).

### 3.5. Identification of the Compounds in AMF Extracts by UPLC-Q-TOF/MS

The dry powder (0.1 g) of ME was re-dissolved in 20 mL of 80% (*v*/*v*) methanol aqueous solution, then the solution was filtered through a 0.22 μm micropore membrane before UPLC-QTOF-MS analysis. Chromatography was performed on an Acquity™ UPLC system (Waters Corp. Milford, MA, USA) equipped with a diode array detector coupled to an electrospray ionization mass detector (UPLC-DAD-ESI/MS^n^). The separation was carried out on an Acquity UPLC^TM^ BEH C18 column (100 mm × 2.1 mm i.d. 1.7 μm; Waters Corp. Milford, MA, USA) maintained at 35 °C. The mobile phase consisted of 0.1% formic acid (HCOOH) in water as solvent A and acetonitrile (ACN) as solvent B, with a gradient elution as follows: 0–9 min, 5–26% B; 9–19 min, 26–65% B; 19–27 min, 65–95% B; 27–27.5 min, 65–95% B; 27.5–30 min, 95–95% B. The flow rate was 0.4 mL/min. The sample injection volume was 3 μL.

Mass detection was performed in positive and negative electrospray modes, using a Synapt^TM^ Q-TOF MS (Waters, Manchester, UK). The gas (N_2_) flows of the cone and desolvation were 50 and 700 L/h, respectively. The temperature of the ion source was maintained at 120 °C and the desolvation temperature was 350 °C. The full scan spectrum was from 100 to 1500 Da. All data collected in centroid mode were acquired and processed using MasslynxNT 4.1 software (Waters Corp. Milford, MA, USA) for peak detection. All analyses were acquired using leucine-enkephalin (ESI^+^: *m*/*z* 556.2771, ESI^−^: *m*/*z* 554.2615) as the lock spray to ensure accuracy and reproducibility.

The compounds were identified by comparing their retention times, UV, and mass spectra with those obtained from reference compounds, when available. Otherwise, compounds were tentatively identified by comparing the obtained information with available data reported in the literature.

### 3.6. Quantification of the Compounds in in AMF Extracts by UPLC-TQ-MS/MS

Analysis was performed on a UPLC-MS/MS system (ACQUITY UPLC, Xevo TQ tandem quadrupole mass spectrometer; Waters Corporation; Milford, MA, United States). About 0.1 g of AMF extract was weighed accurately into a 10 mL volumetric flask and dissolved with 80% methanol to the scale, followed by centrifugation at 4 °C (15,000× *g*, 10 min). The supernatant solution was diluted 10 times with 80% methanol solvent, and the original solution and the diluted sample solution were then filtered through a 0.22 mm filter, respectively. A total of 1 μL of samples was injected into an Acquity UPLC BEH C18 (2.1 mm × 100 mm, 1.7 μm) column using 0.1% formic acid in water (A) and acetonitrile (B) as mobile phase, with the gradient elution as follows: 0–1 min, 5–15% B; 1–14 min, 15–70% B. The flow rate was kept at 0.40 mL/min and the column temperature was maintained at 35 °C throughout the analysis.

The triple quadrupole (TQ) mass spectrometer was operated in both positive and negative modes. The cone voltage and collision energy were optimized for each analyte and selected values are shown in Appendix A, and the UPLC–MS/MS chromatography of 16 markers is presented in Appendix A. The raw data were acquired and processed with MassLynx 4.1 software (Waters Corporation, Milford, MA, USA).

Validation of the method was performed as described in the Appendix A. Concentrations of the target compounds were calculated from the peak areas of the sample and the corresponding standards.

### 3.7. Statistical Analysis

All experiments were carried out in triplicate and results were expressed in means ± standard deviation (SD). One-way analysis of variance (ANOVA) was applied to compare data and the significance of the difference was statistically considered at the level of *p* < 0.05. Relationships between parameters were determined by Pearson’s correlation test. All the analyses were performed with the software of SPSS v. 20. 0 (IBM Corp, Chicago, IL, USA).

## 4. Conclusions

To the best of our knowledge, this work is the first study on total phenols and flavonoids composition combined with the antioxidant activity of AMF. As it was initially proposed, AMF presented the total phenolic and flavonoids profiles, as well as the antioxidant capacity, which may allow these botanical parts to be considered as high value by-products. The ME, a chemical fraction obtained from AMF, exhibited strong antioxidant activity, and was rich in phenolic acids and flavonoids. Overall, the clarification of their phytochemical profile and activity in this research provides useful information for the utilization of the by-product of *Astragalus membranaceus* var. *mongholicus*.

## Figures and Tables

**Figure 1 molecules-24-00434-f001:**
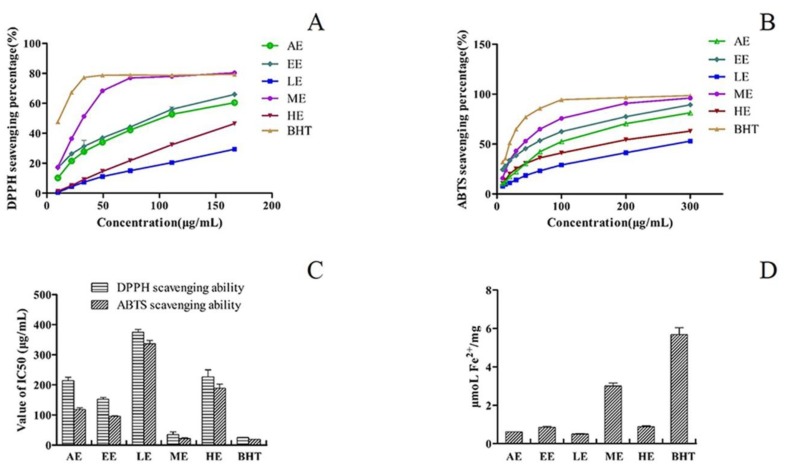
The antioxidant activities and reducing power of different AMF extracts and their fractions. (**A**) DPPH assay; (**B**) ABTS assay; (**C**) IC_50_ values for the radical scavenging activities of different AMF fractions; (**D**) FRAP assay.

**Figure 2 molecules-24-00434-f002:**
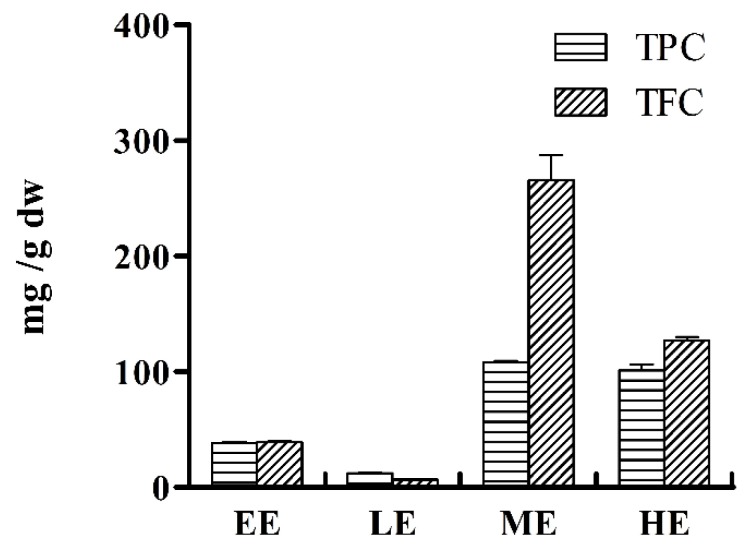
The total phenolic content (TPC) and total flavonoid content (TFC) in different extracts and their fractions of AMF.

**Figure 3 molecules-24-00434-f003:**
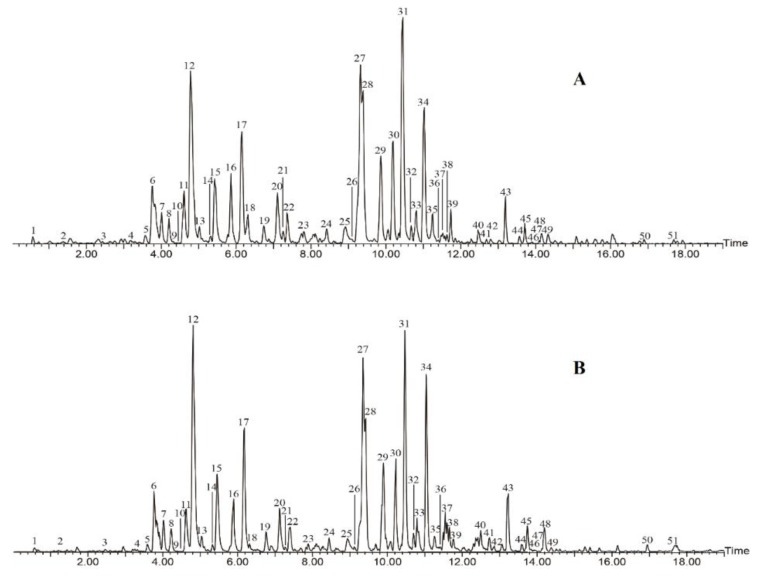
Total ion chromatograms in positive (**A**) and negative (**B**) modes of the extract ME obtained from AMF.

**Table 1 molecules-24-00434-t001:** Compounds identified in ME of AMF by UPLC-Q-TOF MS analysis.

No.	t_R_ (min)	Molecular Formula	[M + H]^+^ (Error, ppm)	Fragment Ions in Positive (+) Ion Mode	[M − H]^−^ (Error, ppm)	Fragment Ions in Negative (−) Ion Mode	λmax (nm)	Identity
1	0.675	C_8_H_8_O_4_	169.0497 [M + H]^+^ (−2.4)		167.0339 [M − H]^−^ (−3.0)		290	vanillic acid ^a^
2	1.371	C_7_H_6_O_4_			153.0192 [M − H]^−^ (2.6)	109.0335 [M − H − CO_2_]^−^	310	protochatechuic acid ^a^
3	2.424	C_9_H_8_O_4_	181.0503 [M + H]^+^ (3.3)		179.0346 [M − H]^−^ (1.1)	135.0482 [M − H − CO_2_]^−^	240, 320	caffeic acid ^a^
4	3.223		927.1859 [M + H]^+^	765.1391 [M + H − 162]^+^ 603.0392 [M + H − 162 − 162]^+^ 465.1044, 301.0693	925.1574	625.1326, 461.1058, 299.0533		Unidentified-*O*-hexoside-hexoside
5	3.585		481.0921 [M + H]^+^	319.0419 [M + H − 162]^+^ 303.0485	479.0791 [M − H]^−^			Unidentified-*O*-hexoside
6	3.766	C_26_H_28_O_16_	597.1466 [M + H]^+^ (1.7)	465.1012 [M + H − 132]^+^ 303.0500 [M + H − 132 − 162]^+^	595.1283 [M − H]^−^ (−2.7)		255, 352	quercetin-*O*-hexoside -pentoside
7	4.011	C_27_H_30_O_16_	611.1594 [M + H]^+^ (−2.9)	465.0987 [M + H − rha]^+^ 303.0524 [M + H − glc − rha]^+^	609.1438 [M − H]^−^ (−3.0)		255, 352	rutin ^a^
8	4.207	C_26_H_28_O_16_	597.1456 [M + H]^+^ (0.0)	465.0993 [M + H − 132]^+^ 303.0482 [M + H − 162 − 132]^+^ 209.1519, 191.1408	595.1277 [M − H]^−^ (−3.7)		254, 351	quercetin-*O*-hexoside -pentoside
9	4.310	C_10_H_10_O_4_			193.0515 [M − H]^−^ (4.1)	178.0311 [M − H − CH_3_]^−^ 134.0392 [M − H − CO_2_ − CH_3_]^−^	320	ferulic acid ^a^
10	4.462	C_22_H_22_O_10_	447.1312 [M + H]^+^ (4.7)	285.0738 [M + H − glc]^+^			260, 290	calycosin-7-*O*-β-d-glucoside ^a^
11	4.614	C_21_H_20_O_12_	465.1029 [M + H]^+^ (−0.9)	303.0492 [M + H − gal]^+^ 287.0528	463.0806 [M − H]^−^ (−3.2)		256, 352	hyperoside ^a^
12	4.780	C_21_H_20_O_12_	465.1020 [M + H]^+^ (−2.8)	303.0477 [M + H − glc]^+^	463.0867 [M − H]^−^ (−2.2)	301.0321 [M − H − glc]^−^	255, 352	isoquercitrin ^a^
13	5.020	C_28_H_32_O_16_	625.1775 [M + H]^+^ (1.0)	647.1636 [M + Na]^+^ 479.1188 [M + H − rha]^+^ 317.0630 [M + H − glc − rha]^+^ 303.0475, 197.1146	623.1603 [M − H]^−^ (−1.4)	609.1362 [M − CH_3_]^−^ 463.0808 [M − CH_3_ − rha]^−^		isorhamnetin 3-*O*-rutinoside
14	5.324		941.1999 [M + H]^+^	779.1476 [M + H − 162]^+^; 617.1019 [M + H − 162 − 162]^+^ 697.5236, 479.1232, 317.0638, 303.0507				isorhamnetin-*O*-hexoside-hexoside derivative
15	5.427	C_24_H_22_O_15_	551.1027 [M + H]^+^ (−1.8)	303.0479, 287.0526, 273.0385	549.0860 [M − H]^−^ (−3.6)	505.0948 [M − H − CO_2_]^−^ 301.0316	255, 352	quercetin-*O*-malonyl-hexoside
16	5.862	C_21_H_20_O_11_	449.1079 [M + H]^+^ (−1.1)	287.0550 [M + H − glc]^+^, 229.0483	447.0943 [M − H]^−^ (3.1)	285.0388 [M − H − glc]^−^	265, 346	astragalin ^a^
17	6.152	C_22_H_22_O_12_	479.1201 [M + H]^+^ (2.3)	317.0593 [M + H − glc]^+^	477.1026 [M − H]^−^ (−1.5)	315.0468 [M − H − glc]^−^	254, 352	isorhamnetin-3-*O*-β-d-glucoside ^a^
18	6.304		679.5200 [M + H]^+^	701.4979 [M + Na]^+^ 566.4278, 453.3440, 317.0627, 303.0491, 287.0552, 273.0742	677.4871 [M − H]^−^	723.4912 [M + HCOO]^−^ 477.0983		Unidentified
19	6.740		535.1082 [M + H]^+^	557.0948 [M + Na]^+^ 517.0905 [M + H − H_2_O]^+^ 462.3415	533.0818 [M − H]^−^	489.0993 [M − H − CO_2_]^−^ 285.0410	265, 344	flavonoid
20	7.117		792.5888 [M + H]^+^	814.5846 [M + Na]^+^, 679.5115, 565.1169, 396.7961, 317.0635	790.5735 [M − H]^−^	836.5712 [M + HCOO]^−^ 519.1112, 315.0487	254, 354	flavonoid
21	7.254	C_22_H_22_O_11_	463.1234 [M + H]^+^ (−1.3)	301.0672 [M + H − glc]^+^ 286.1731 [M + H − glc − CH_3_]^+^	461.1041 [M − H]^−^ (−2.8)	299.0568 [M − H − glc]^−^	238, 266, 332	pratensein-7-*O*-β-d-glucoside
22	7.371	C_33_H_40_O_21_	773.2129 [M + H]^+^ (−1.4)	795.1703 [M + Na]^+^ 611.6411 [M + H − 162]^+^ 465.1052 [M + H − 162 − 146]^+^ 303.0500 [M + H − 162 − 146 − 162]^+^	771.1979W [M − H]^−^ (−0.6)		254, 356	quercetin-*O*-rutinoside-hexoside
23	7.812		905.6908 [M + H]^+^	927.6686 [M + Na]^+^, 679.5251, 566.4298, 453.3409, 341.2460	903.6592 [M − H]^−^	949.6523 [M + HCOO]^−^ 813.4478, 519.1101, 343.2118		Unidentified
24	8.370		1018.7797 [M + H]^+^	1040.7581 [M + Na]^+^ 509.8839, 341.2454 288.1595		1062.7168, 741.1525 623.1522, 515.2469 343.2078		Unidentified
25	8.929	C_15_H_10_O_7_	303.0501 [M + H]^+^ (−1.3)	285.0747 [M + H − H_2_O]^+^	301.0347 [M − H]^−^ (−0.3)	283.0600 [M − H − H_2_O]^−^	254, 366	quercetin ^a^
26	9.061	C_16_H_12_O_5_	285.0756 [M + H]^+^ (−2.5)	270.0500 [M + H − CH_3_]^+^			250, 190	calycosin ^a^
27	9.328	C_22_H_22_O_12_	479.1185 [M + H]^+^ (−1.0)	317.0656 [M + H − 162]^+^ 299.0548, 183.0756, 163.0363	477.1028 [M − H]^−^ (−1.0)	315.0489 [M − H − 162]^−^	258, 350	isorhamnetin-*O*-hexoside
28	9.380	C_23_H_26_O_10_	463.1592 [M + H]^+^ (−2.6)	301.0672 [M + H − glc]^+^			256, 354	(−)-methylinissolin-3-*O*-β-d-glucoside ^a^
29	9.857		565.1182 [M + H]^+^	419.3312 (146) 317.0627, 243.0627	563.0993 [M − H]^−^	519.1140 [M − H − CO_2_]^−^	255, 353	Unidentified-*O*-rhamnoside
30	10.186	C_31_H_34_O_18_	695.1833 [M + H]^+^ (1.4)	717.1677 [M + Na]^+^ 549.1226 [M + H − 146]^+^ 463.1245 [M + H − 146 − 86]^+^, 301.0694 [M + H − 146 − 86 − 162]^+^	693.1659 [M − H]^−^ (1.2)	649.1680 [M − H − CO_2_]^−^ 461.1049 [M − H − 146 − 86]^−^	266, 347	rhamnocitrin-*O*-malonyl-glucoside-rhamnoside
31	10.438	C_22_H_22_O_11_	463.1233 [M + H]^+^ (−1.5)	485.1034 [M + Na]^+^ 301.0672 [M + H − 162]^+^	461.1075 [M − H]^−^ (−2.0)	299.0542 [M − H − glc]^−^	266, 347	rhamnocitrin-*O*-hexoside
32	10.678		633.3999 [M + H]^+^	655.1724 [M + Na]^+^ 615.3941 [M + H − H_2_O]^+^; 453.3354 [M + H − 162 − H_2_O]^+^ 597.3831, 435.3251, 301.0700				Unidentified-*O*-hexoside
33	10.947	C_15_H_10_O_6_	287.0549 [M + H]^+^ (−2.4)		285.0395 [M − H]^−^ (−1.4)			kaempferol ^a^
34	11.018	C_25_H_24_O_14_	549.1242 [M+H]^+^ (−0.4)	301.0688 [M + H − 162 − 86]^+^ 167.0326	547.1067 [M − H]^−^ (−3.8)	503.1147 [M − H − CO_2_]^−^ 299.0634 [M − H − 162 − 86]^−^	266, 347	rhamnocitrin-*O*-malonyl-hexoside
35	11.229	C_25_H_28_O_11_	505.1710 [M + H]^+^ (0.0)	301.0688 [M + H − glc − 42]^+^ 286.0468, 167.0296	503.1541 [M − H]^−^ (−2.4)		265, 346	9,10-diMP-3-*O*-acetyl-glucoside
36	11.295	C_16_H_12_O_7_	317.0658 [M + H]^+^ (−0.9)	287.0431	315.0609 [M − H]^−^ (1.3)	300.0350		isorhamnetin
37	11.520		797.4355 [M + H]^+^	819.4246 [M + Na]^+^ 647.3790, 629.3679, 453.3352, 301.0690	795.4099 [M − H]^−^	699.4210		Unidentified-*O*-penoside
38	11.643	C_43_H_70_O_15_	827.4764 [M + H]^+^ (−3.5)	849.4391 [M + Na]^+^ 647.3793 [M + H − glc]^+^ 629.3701 [M + H − glc − H_2_O]^+^	825.4608 [M − H]^−^ (−3.4)	697.4040 327.2169		isoastragaloside II ^a^
39	11.765		437.3441 [M + H]^+^	455.3521 [M + Na]^+^ 419.3271, 401.3217, 301.0702				Unidentified
40	12.490		637.4607 [M + H]^+^	659.4114 [M + Na]^+^ 601.4108, 421.3460, 249.1829	635.4064 [M − H]^−^	681.4106 [M + HCOO]^−^ 329.2305		Unidentified
41	12.701		795.4209 [M + H]^+^	817.4014 [M + Na]^+^ 627.3550, 609.3422, 469.3301, 451.3196, 343.1530, 311.2187 253.2164, 217.1928	793.3873 [M − H]^−^	287.2206		Unidentified
42	12.818		825.4285 [M + H]^+^	847.4200 [M + Na]^+^ 627.3531, 609.3378, 469.3327, 317.0642	823.3981 [M − H]^−^	677.3810 [M − H − 146]^−^ 631.3703, 315.0492		Unidentified-*O*-rhamnoside
43	13.181		695 [M + H]^+^	717.4201 [M + Na]^+^ 659.4115, 497.3633, 479.3441	693.4111 [M − H]^−^	739.4180 [M + HCOO]^−^		Unidentified
44	13.558		635 [M + H]^+^	657 [M + Na]^+^ 617.4112, 599.3976, 437.3422, 419.3292, 401.3216	633.3881 [M − H]^−^	679.3937 [M + HCOO]^−^		Unidentified
45	13.710		819.3751 [M + H]^+^	837.3528 [M + Na]^+^ 798.4721, 745.4219, 727.4098, 685.3970, 667.3876, 497.3634				Unidentified
46	13.906		957.5087 [M + H]^+^	979.4908 [M + Na]^+^ 811.4536 [M + H − 146]^+^				Unidentified-*O*-rhamnoside
47	14.130	C_46_H_72_O_18_	913.4787 [M + H]^+^ (−1.1)	935.5103 [M + Na]^+^ 895.5112 [M + H − H_2_O]^+^	911.4661 [M − H]^−^ (2.3)			malonylastragaloside II isomer
48	14.160	C_48_H_78_O_18_	943.5286 [M + H]^+^ (2.1)	965.5129 [M + Na]^+^ 797.4725 [M + H − rha]^+^ 635.4149 [M + H − glc − rha]^+^ 617.4098 [M + H − glc − rha − H_2_O]^+^ 599.3946 [M + H − glc − rha − 2H_2_O]^+^ 441.3738, 423.3643, 405.3457, 203.1721	941.5108 [M − H]^−^ (−0.2)			soyasaponin I ^a^
49	14.327	C_16_H_12_O_6_	301.0706 [M + H]^+^ (−2.0)		299.0549 [M − H]^−^ (−2.3)	284.0317 [M − H − CH_3_]^−^	228	rhamnocitrin ^a^
50	16.908		781.4708 [M + H]^+^	803.4598 [M + Na]^+^ 763.4578 [M + H − H_2_O]^+^ 601.4130 [M + H − 162 − 18]^+^	779.4440 [M − H]^−^			Unidentified-*O*-hexoside
51	17.707		619.4214 [M + H]^+^	641.4061 [M + Na]^+^ 601.4132 [M + H − H_2_O]^+^ 583.3962 [M + H − 2H_2_O]^+^ 439.3566 [M + H − 162 − H_2_O]^+^; 421.343 [M + H − 162 − 2H_2_O]^+^ 403.3362	617.3951 [M − H]^−^	663.3966 [M + HCOO]^−^ 265.1462		Unidentified-*O*-hexoside

^a^ Compared with standard compound.

**Table 2 molecules-24-00434-t002:** The contents (mg/g) of 16 investigated compounds in extracts of AMF.

Analytes	Samples				
	AE	EE	LE	ME	HE
protochatechuic acid	0.169 ± 0.026	0.390 ± 0.093	0.385 ± 0.004	0.322 ± 0.015	nd ^a^
caffeic acid	nd	0.053 ± 0.008	nd	0.226 ± 0.004	nd
vanillic acid	0.134 ± 0.004	0.267 ± 0.039	nd	0.755 ± 0.008	0.017 ± 0.001
ferulic acid	0.064 ± 0.001	0.176 ± 0.024	nd	0.568 ± 0.015	nd
rutin	0.663 ± 0.021	1.952 ± 0.162	nd	6.099 ± 0.080	0.018 ± 0.001
calycosin-7-*O*-β-d-glucoside	nd	0.024 ± 0.002	nd	0.048 ± 0.001	nd
hyperoside	4.669 ± 0.510	6.574 ± 0.292	nd	16.285 ± 0.195	nd
astragalin	1.063 ± 0.073	1.674 ± 0.026	nd	4.810 ± 0.028	0.042 ± 0.002
isorhamnetin 3-*O*-β-d-glucoside	1.673 ± 0.055	2.326 ± 0.128	0.0139 ± 0.001	4.970 ± 0.048	0.079 ± 0.003
(−)-methylinissolin 3-*O*-β-d-glucoside	0.219 ± 0.016	0.322 ± 0.005	nd	0.838 ± 0.027	0.044 ± 0.002
quercetin	0.133 ± 0.017	1.159 ± 0.029	0.018 ± 0.001	2.295 ± 0.038	1.326 ± 0.055
calycosin	nd	0.095 ± 0.006	nd	0.086 ± 0.002	4.558 ± 0.073
kaempferol	nd	0.141 ± 0.016	0.025 ± 0.001	0.088 ± 0.002	0.562 ± 0.017
isorhamnetin	0.009 ± 0.001	0.142 ± 0.006	nd	0.042 ± 0.001	0.551 ± 0.006
formononetin	nd	0.020 ± 0.001	nd	nd	0.110 ± 0.001
rhamnocitrin	0.022 ± 0.005	1.425 ± 0.096	nd	0.224 ± 0.008	2.184 ± 0.053
Total content of phenolic acids	0.367 ± 0.011	0.886 ± 0.058	0.385 ± 0.012	1.871 ± 0.026	0.017 ± 0.002
Total content of flavonoids	8.451 ± 0.523	16.973 ± 0.854	0.057 ± 0.004	36.399 ± 1.230	10.801 ± 0.452

^a^ not detected.

**Table 3 molecules-24-00434-t003:** Pearson’s correlation coefficients of antioxidant activities with phenolic acids and flavonoids contents for AMF.

Analytes	DPPH	ABTS	FRAP
TPC ^a^	0.658 *	0.713 **	0.805 **
TFC	0.717 **	0.750 **	0.948 **
protochatechuic acid	0.181	0.320	0.514
caffeic acid	0.785	0.776	0.980 **
vanillic acid	0.691 *	0.691 *	0.953 **
rutin	0.684 *	0.677 *	0.946 **
calycosin-7-*O*-β-d-glucoside	0.773	0.756	0.974 **
hyperoside	0.737 *	0.678 *	0.960 **
ferulic acid	0.739 *	0.684 *	0.963 **
astragalin	0.679 *	0.669 *	0.944 **
isorhamnetin-3-*O*-β-d-glucoside	0.672 **	0.660 **	0.943 **
(−)-methylinissolin-3-*O*-β-d-glucoside	0.675 *	0.670 *	0.949 **
quercetin	0.575 *	0.604 *	0.872 **
calycosin	−0.278	−0.189	−0.186
kaempferol	0.022	0.097	0.067
isorhamnetin	−0.190	−0.048	−0.027
formononetin	−0.018	−0.074	0.338
rhamnocitrin	−0.168	−0.017	−0.004

^a^ TPC, TFC: total phenols and flavonoids contents, respectively; DPPH, ABTS: DPPH radical scavenging activity at 166.7 μg/g, 74.1 μg/g, and 32.9 μg/g, and ABTS radical scavenging activity at 100 μg/g, 44.4 μg/g, and 29.8 μg/g, respectively; FRAP: ferric reducing antioxidant power was expressed as μM Fe^2+^ equivalent expression of 2 mg, 1 mg, and 0.4 mg dry extract, respectively. * Significant at *p* < 0.05. ** Significant at *p* < 0.03.

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
