# Peer review of "Flowers of *Astragalus membranaceus* var. *mongholicus* as a Novel High Potential By-Product: Phytochemical Characterization and Antioxidant Activity"

_molecules, 2019, doi:10.3390/molecules24030434_

Round 1

Reviewer 1 Report

The manuscript “ Flowers of Astragalus membranaceus var. mongholicus as a novel high potential by-product: Phytochemical characterization and antioxidant activity” by shows a solid and well done experimental work. The aims of the paper is clearly defined and the introduction can be considered updated. The methodology is correct and appropriate, and the results are consistent. The materials and methods are clearly defined and allow a perspective of accurracy/precision of data obtained by the authors. The discussion is supported by data. The conclusion is concise and objective. I think the article is fine, but needs some corrections as noted below.

The paper, according to my perspective deserves publication after these revisions:

1)      A major question regard the quantification of total phenols and of total flavonoids. From the discussion  and experimental section, it is not clear what the authors determined: the total phenolics content, as reported in the text, or the total phenolic acids, as reported in fig. 2? In the first case, how is the TPC lower than the TFC possible? Flavonoids are polyphenols and they do react with the Folin-Ciocalteu reagent. How are the phenolic acids estimated?

2)      Title: the terms Astragalus membranaceus, and mongholicus should be written in Italic

Results and discussion section: a) the title of paragraph 2.1 should be moved to line 70; thus, the title of paragraph 2.2, missing, should be added to line 92.

Author Response

Response to Reviewer 1 Comments

Point 1: A major question regard the quantification of total phenols and of total flavonoids. From the discussion and experimental section, it is not clear what the authors determined: the total phenolics content, as reported in the text, or the total phenolic acids, as reported in fig. 2 ? In the first case, how is the TPC lower than the TFC possible? Flavonoids are polyphenols and they do react with the Folin-Ciocalteu reagent. How are the phenolic acids estimated?

Response 1: Thank you a lot for pointing out this problem. We are sorry that the description of the total phenolic acid content in the article is not appropriate, and in this regard, we have made a modification in the article, which we refer to not the total phenolic acid content but the total phenolics content. In addition, since the determination method and the selected reference substance of the total flavonoids and the total phenolics contents are different, for example, the content of total flavonoids in this paperQawasmeh A , Obied H K , Raman A , et al. Influence of Fungal Endophyte Infection on Phenolic Content and Antioxidant Activity in Grasses: Interaction between Lolium perenne and Different Strains of Neotyphodium lolii. Journal of Agricultural and Food Chemistry, 2012, 60(13):3381-3388. is also higher than that of total phenols. We think that the content of the total flavonoids and the total phenolics can’t be compared.

Point 2: Title: the terms Astragalus membranaceus, and mongholicus should be written in Italic Results and discussion section: a) the title of paragraph 2.1 should be moved to line 70; thus, the title of paragraph 2.2, missing, should be added to line 92.

Response 2: Thank you for your instructive suggestions. We have modified the question of Latin italics in the title, moved the 2.1 title and added the 2.2 title, and marked it in red.

Reviewer 2 Report

Dear Authors,

I read the manuscript " Flowers of Astragalus membranaceus var. mongholicus as a novel high potential by-product: Phytochemical characterization and antioxidant activity”. The manuscript deals about an interesting topic, providing a very impressive sequence of data about compound structure determination. Some more questions or notes are here reported.

-In the manuscript, the first time you use an acronym, you should explain what it means. So, for instance, it would be better to specify that ME is an extract the first time you present it, writing: “ ME extract” or“ME (see Material and Method)”. 

-Figure 3: there are more peaks than those numbered. If those peaks are impurity not always present it should be reported somewhere in the presentation of HPLC analyses, if not, even those peaks should be presented and the structure determined.

Furthermore, I would suggest some minor changes in phrases, as reported here after:

Line 19 ” its flowers (AMF) were often…” instead of “it flower (AMF) was often”

Line 27 “31 total compounds” instead “Total 31 compounds”

Line 52 “Besides, flavonoids, sugars, amino acids and triterpenoid saponins were definitely detected in the flower of A. membranaceus instead “Besides, the flavonoids, sugars, amino acids and triterpenoid saponins 52 was definitely detected in the flower of A. membranaceus”

Line 60 “Chinese tonic medicine” instead “tonic Chinese medicine” or eventually “tonic of Chinese medicine”

Line “the ability of AMF extracts to scavenge free ABTS and DPPH radicals showed a growing trend with increasing sample concentration” instead of “the scavenging ability of AMF extracts to DPPH and ABTS free radicals showed an increasing trend with the increase of sample concentration”.

Line 85 the phrase is not clear may be it lacks a verb

Line 94 “relationship” or” correlation“ instead  of “relativity”

Line 100 May be you intend: “indicating that there could be a positive correlation between them.”

Line 106 “owing to its best results” instead “owing its best results”.

Line 148: “peak 15 tentatively assigned as quercetin-O-malonyl-hexoside. Instead of “peak 15 tentatively assigned as and quercetin-O-malonyl-hexoside”.

Line 149: erase “were

Line 196 “In addition, m/z 647.3793” instead “In addition, 647.3793”

Line 243-245: the phrase is not clear, please explain the concept in a more understandable way.

Line 259-60 Is not clearer “Strong relationships (r 0.658 to 0.948) [28] were observed between the antioxidant capacity (DPPH, ABTS and FRAP) and the TPC and TFC in AMF extracts” than “The strong relationships (r from 0.658 to 0.948) [28]were observed in antioxidant capacities (DPPH, ABTS and FRAP) as well as TPC and TFC in the extracts of AMF”?

Line 392: erase “composition”

Line 374-5: Check Supplementary Material because there are two Table S1 and no Table S3.

For all this reason, I do believe that this manuscript deserves to be published in Molecules after minor revisions.

Author Response

Point 1: In the manuscript, the first time you use an acronym, you should explain what it means. So, for instance, it would be better to specify that ME is an extract the first time you present it, writing: “ME extract ” or “ ME (see Material and Method) ”

Response 1: Thank you for your careful work. In the manuscript, we have added the meaning of acronyms to the first use of acronyms.

Point 2: Figure 3: there are more peaks than those numbered. If those peaks are impurity not always present it should be reported somewhere in the presentation of HPLC analyses, if not, even those peaks should be presented and the structure determined.

Response 2: Thank you for your instructive suggestions. According to your comments, we have added some text describing other unnumbered peaks. The revised content is showed in lines 114-117.

Point 3: Line 19 “its flowers (AMF) were often…” instead of “it flower (AMF) was often”

Response 3: Thank you for your careful work. We have corrected the words in the revised manuscript.

Point 4: Line 27 “31 total compounds” instead “Total 31 compounds”

Response 4: Thank you for your careful work. We have corrected the words in the revised manuscript.

Point 5: Line 52 “Besides, flavonoids, sugars, amino acids and triterpenoid saponins were definitely detected in the flower of A. membranaceus instead “Besides, the flavonoids, sugars, amino acids and triterpenoid saponins 52 was definitely detected in the flower of A. membranaceus

Response 5: Thank you for your careful work. We have corrected the words in the revised manuscript.

Point 6: Line 60 “Chinese tonic medicine” instead “tonic Chinese medicine” or eventually “tonic of Chinese medicine”

Response 6: Thank you for your careful work. We have corrected the words in the revised manuscript.

Point 7: Line 76 “the ability of AMF extracts to scavenge free ABTS and DPPH radicals showed a growing trend with increasing sample concentration” instead of “the scavenging ability of AMF extracts to DPPH and ABTS free radicals showed an increasing trend with the increase of sample concentration”.

Response 7: Thank you for your careful work. We have corrected the words in the revised manuscript.

Point 8: Line 85 the phrase is not clear may be it lacks a verb

Response 8: Thank you for pointing out this problem. We are sorry for this language mistake that made the meaning of this sentence unclear. We have corrected the words in the revised manuscript.

Point 9: Line 94 “relationship” or” correlation” instead of “relativity”

Response 9: Thank you for your careful review of this paper, we have corrected the words in the revised manuscript.

Point 10: Line 100 May be you intend: “indicating that there could be a positive correlation between them.”

Response 10: Thanks for your instructive advice. We have modified it in the revision.

Point 11: Line 106 “owing to its best results” instead “owing its best results”.

Response 11: Thanks a lot for your careful guidance. We have corrected the

corresponding writing.

Point 12: Line 148: “peak 15 tentatively assigned as quercetin-O-malonyl-hexoside. Instead of “peak 15 tentatively assigned as and quercetin-O-malonyl-hexoside”.

Response 12: Thank you for putting forward some details. According to the comments, we have made several modifications in the revised manuscript

Point 13: Line 149: erase “were”.

Response 13: Thank you for your careful work. We have corrected the words in the

revised manuscript.

Point 14: Line 196 “In addition, m/z 647.3793” instead “In addition, 647.3793”

Response 14: Thanks for your instructive advice. We have modified it in the revision.

Point 15: Line 243-245: the phrase is not clear, please explain the concept in a more understandable way.

Response 15: Thank you for pointing out this problem. We are sorry for this language mistake that made the meaning of this sentence unclear, and we have revised the sentence.

Point 16: Line 259-60 Is not clearer “Strong relationships (r 0.658 to 0.948) [28] were observed between the antioxidant capacity (DPPH, ABTS and FRAP) and the TPC and TFC in AMF extracts” than “The strong relationships (r from 0.658 to 0.948) [28] were observed in antioxidant capacities (DPPH, ABTS and FRAP) as well as TPC and TFC in the extracts of AMF”?

Response 16: Thank you for your careful review of this paper, we are sorry for this sentence unclear. We have adjusted it in the revised manuscript, according to your suggestion.

Point 17: Line 392: erase “composition”

Response 17: Thanks for your instructive advice. We have modified it in the revision.

Point 18: Line 374-5: Check Supplementary Material because there are two Table S1 and no Table S3.

Response 18: Thank you for pointing out this problem. We carefully checked the header part of the supplementary material and modified the number of Table S3.

Reviewer 3 Report

The work is interesting, well-constructed and written, but requires some additions and corrections.

The abbreviations (EE, AE and so on) should be explained in the place of their first use, otherwise the text is difficult to read.

Line 43-44: “Hence, the bulk of the commercial supply is taken from farming sources in China [1].” This fragment should not be quoted after Fu et al, 2014, because in the article by Fu et al. information on the origin of the raw material is cited after Ma QX et. al, 2000 (unfortunately it is not available on the internet). In addition, this sentence is prescribed literally from the article by Fu et al., which should not take place.

I have been working on A. membranaceus for many years and so far I have not found any information on the removal of flowers of this plant as a procedure to increase the yield of roots. Thus I suggest to quote a scientific or popular science work confirming this practice. In addition, this activity (removal of flowers) seems to be extremely labor-intensive, so is it really performed on plantations? is it really profitable?

In the methodology, there is no description of the raw material used for research (only 1 sentence on this topic - line 279-280). Please write exactly how the raw material was collected. Whether it came from wild growing plants or from cultivation. If from cultivation then at what age were the plants. How was the raw material treated after harvesting. The temperature, time, length and other drying conditions are crucial in the case of medicinal plants in determining the composition and content of active compounds. The same raw material dried in different conditions can be diametrically different in this respect.

How the raw material was stored until analysis (length of storage period, temperature, light access).

How the extracts were obtained – the drying method is not described (apparatus and drying conditions should be added). What was the water content in finally obtained dry extracts.

Some fragments should be verified in terms of language (e.g. Line 96-94).

Line 89: Figure titles: The antioxidant activities and reducing power of different AMF fractions

In my opinion it should be: The antioxidant activities and reducing power of different AMF extracts and their fractions (the same fig. 2).

Author Response

Reviewer 3:

Point 1: The abbreviations (EE, AE and so on) should be explained in the place of their first use, otherwise the text is difficult to read.

Response 1: Special thanks for your careful review. It is our carelessness and we are sorry about this. According to your comments, related content have been improved. We have added the meaning of acronyms to the first use of acronyms.

Point 2: Line 43-44: “Hence, the bulk of the commercial supply is taken from farming sources in China [1].” This fragment should not be quoted after Fu et al, 2014, because in the article by Fu et al. information on the origin of the raw material is cited after Ma QX et. al, 2000 (unfortunately it is not available on the internet). In addition, this sentence is prescribed literally from the article by Fu et al., which should not take place.

Response 2: Thank you for pointing out this problem. We apologize for our inappropriate language and improper use of the literature. We have revised the sentence and re-cited the reference.

Point 3: I have been working on A. membranaceus for many years and so far I have not found any information on the removal of flowers of this plant as a procedure to increase the yield of roots. Thus I suggest to quote a scientific or popular science work confirming this practice. In addition, this activity (removal of flowers) seems to be extremely labor-intensive, so is it really performed on plantations? Is it really profitable?

Response 3: Thank you for your careful work. We have consulted a lot of literatures on AMF, it is reported here that the yield of Astragali Radix can be increased by removing the top inflorescences in the process of A. membranaceus var. monghoicus cultivation, and the references have been cited in the manuscript to confirm the practice. In addition, we also found in the field investigation that the top inflorescences will indeed be removed at the end of July to improve the yield of Astragali Radix in some cultivation regions of Gansu and Ningxia provinces. Besides, as far as we known, in the northeast of China, for example, the Daxing'an Mountain range area, the flowers of A. membranaceus has been used as the scented tea and vegetable, and the research results of our research group showed that AMF is rich in monosaccharide, polysaccharide component and amino acids. Hence, the aim of our research lies in the discovery of the utilization value of AMF, as a by-product of Astragalus membranaceus, so that we can make use of the wastes in the process of Astragali Radix cultivation and bring more benefits to us.

Point 4: In the methodology, there is no description of the raw material used for research (only 1 sentence on this topic - line 279-280). Please write exactly how the raw material was collected. Whether it came from wild growing plants or from cultivation. If from cultivation then at what age were the plants. How was the raw material treated after harvesting. The temperature, time, length and other drying conditions are crucial in the case of medicinal plants in determining the composition and content of active compounds. The same raw material dried in different conditions can be diametrically different in this respect.

Response 4: Thank you for putting forward some details. According to the comments, we have added a description of the collection of raw materials and the pre-harvest processing process in lines 282-285.

Point 5: How the raw material was stored until analysis (length of storage period, temperature, light access).

Response 5: Thanks for your comment. Immediately following the fourth point, we described the stock conditions of the raw materials in lines 282-285.

Point 6: How the extracts were obtained – the drying method is not described (apparatus and drying conditions should be added). What was the water content in finally obtained dry extracts.

in lines 305-309 with red words.

Point 7: Some fragments should be verified in terms of language (e.g. Line 96-94).

Response 7: Thank you for pointing out this problem. We have carefully revised the sentence in the revised manuscript.

Point 8: Line 89: Figure titles: The antioxidant activities and reducing power of different AMF fractions, in my opinion it should be: The antioxidant activities and reducing power of different AMF extracts and their fractions (the same fig. 2).

Response 8: Thank you for your careful work. We have corrected the words in tables of the figure revised manuscript.

Round 2

Reviewer 1 Report

The manuscript has been substantially improved. The authors answered all my concerns adequately. The revised manuscript is recommended for publication.